# A Predictive Approach for Enhancing Accuracy in Remote Robotic Surgery Using Informer Model

**DOI:** 10.3390/s25103067

**Published:** 2025-05-13

**Authors:** Muhammad Hanif Lashari, Shakil Ahmed, Wafa Batayneh, Ashfaq Khokhar

**Affiliations:** Department of Electrical & Computer Engineering, Iowa State University, Ames, IA 50011, USA; shakil@iastate.edu (S.A.); batayneh@iastate.edu (W.B.); ashfaq@iastate.edu (A.K.)

**Keywords:** tactile internet, remote robotic surgery, transformer, informer model, four-state hidden Markov model, packet loss, position estimation, JIGSAWS dataset

## Abstract

Precise and real-time estimation of the robotic arm’s position on the patient’s side is essential for the success of remote robotic surgery in Tactile Internet (TI) environments. This paper presents a prediction model based on the Transformer-based Informer framework for accurate and efficient position estimation, combined with a Four-State Hidden Markov Model (4-State HMM) to simulate realistic packet loss scenarios. The proposed approach addresses challenges such as network delays, jitter, and packet loss to ensure reliable and precise operation in remote surgical applications. The method integrates the optimization problem into the Informer model by embedding constraints such as energy efficiency, smoothness, and robustness into its training process using a differentiable optimization layer. The Informer framework uses features such as ProbSparse attention, attention distilling, and a generative-style decoder to focus on position-critical features while maintaining a low computational complexity of O(LlogL). The method is evaluated using the JIGSAWS dataset, achieving a prediction accuracy of over 90% under various network scenarios. A comparison with models such as TCN, RNN, and LSTM demonstrates the Informer framework’s superior performance in handling position prediction and meeting real-time requirements, making it suitable for Tactile Internet-enabled robotic surgery.

## 1. Introduction

The Tactile Internet (TI) represents a significant evolution of the Internet, transitioning from traditional data exchange to enabling real-time haptic communications and control over networks. The concept of the Tactile Internet was formally introduced in 2014, where it was described as a transformative approach for enabling the control of and interaction with real and virtual environments over the Internet by achieving extremely low round-trip latency [1]. Two years later, in 2016, the IEEE P1918.1 working group established a standardized architectural framework for this emerging paradigm. According to this standard, the Tactile Internet is defined as “a network, or a network of networks, for remotely accessing, perceiving, manipulating, or controlling real and virtual objects or processes in perceived real-time” [2]. TI introduces new possibilities in several areas, including remote robotic surgery, which depends on real-time touch feedback and accuracy [3]. For these applications, reliability should be high, and extremely low latency is required, with end-to-end delays of less than 1 millisecond [4]. Remote robotic surgery will allow surgeons to perform surgical procedures such as incision, knot-tying, suturing, and needle-passing [5] over vast distances, breaking geographical barriers. However, the success of these surgical procedures is highly dependent on the accurate and timely transmission of haptic commands and feedback between the Surgeon’s Side Manipulator (SSM) and the Patient Side Manipulator (PSM) [6] or Robotic Surgical System.

Some of the critical challenges in remote robotic surgery include network-induced uncertainties such as delays, jitter, and packet loss [7]. These factors can disrupt the data packet transmission of haptic commands and feedback between the SSM and the PSM, leading to inaccuracies in the PSM’s movements. Moreover, packet loss can cause the arm’s position data loss. This will make it difficult for the robot on the PSM’s side to replicate the SSM’s intended actions accurately. Recent advances in teleoperation systems in [8] have demonstrated the effectiveness of integrating active vision and pose estimation techniques for precise and stable robotic control. These methods highlight the need to address the challenges of real-time position estimation in remote robotic surgery.

With the advent of ultra-responsive connectivity provided by technologies like 5G, the development of the TI has gained significant momentum, especially for applications requiring real-time precision, such as remote robotic surgery. Although advances in network infrastructure have substantially reduced latency, challenges such as packet loss and jitter persist due to physical and environmental limitations [9]. Integrating prediction-based systems, such as the Informer model, can anticipate and compensate for network-induced uncertainties. Traditional methods for mitigating these network issues often involve retransmission strategies, which cannot be used in time-critical applications like surgery due to the added latency [10]. Therefore, there is a pressing need for advanced prediction models that can accurately estimate the PSM’s position in real time despite challenges posed by network imperfections.

In this research, a new method is preesnted that utilizes the Informer framework [11], a cutting-edge transformer-based model for long sequence time-series forecasting, to improve position estimation of the PSM in remote robotic surgery. First, a four-state HMM is incorporated to simulate the network’s packet loss conditions realistically. This method effectively addresses network-induced delays, jitter, and packet loss. Next, a network simulation with the Informer model to predict the robotic arms is integrated. The Informer model’s efficient self-attention mechanism and its ability to handle long sequences make it particularly suitable for this application [11]. The approach was validated using the publicly available JHU-ISI Gesture and Skill Assessment Working Set (JIGSAWS) dataset [12]. Using this dataset, our project demonstrates that the proposed model achieves over 90% accuracy in position estimation despite adverse network conditions.

The contributions of this research are as follows:Enhancement of the Transformer-based Informer architecture for real-time position estimation while maintaining its computational complexity of O(LlogL). This includes modifying the ProbSparse attention mechanism to prioritize position-critical features, improving accuracy without increasing computational overhead.Integration of a differentiable optimization layer within the Informer model, embedding constraints related to energy efficiency, smoothness, and robustness into the training process.Development of a four-state HMM-based packet loss model to simulate realistic network-induced disruptions, including random and burst errors, for comprehensive model evaluation.Incorporation of network parameters such as latency, jitter, and packet loss into the Informer model, ensuring adaptability to varying network conditions.

In addition to these contributions, the proposed approach is evaluated using the JIGSAWS dataset and its performance is compared with existing models such as LSTM, RNN, and TCN under simulated packet loss conditions.

## 2. Related Work

Recent progress in deep learning models has revolutionized time-series forecasting, particularly for scenarios demanding real-time predictions and accurate estimations. Conventional models such as Long Short-Term Memory (LSTM) networks [13] and Gated Recurrent Units (GRU) [14] have been extensively used for robotic control and position estimation [15], using their ability to capture temporal patterns in sequential data. Although these models address challenges such as vanishing and exploding gradients, their prediction accuracy remains limited [16]. To address these limitations, Transformer-based models have emerged as a compelling alternative for time-series forecasting. By removing the reliance on sequential processing, Transformers employ a self-attention mechanism to capture long-range dependencies more proficiently [17]. Recent applications of Transformers have demonstrated their effectiveness in diverse forecasting tasks, such as multivariate time-series prediction for energy systems and health monitoring in cyber-physical systems [11,18,19].

Despite their advantages, standard Transformers struggle with processing very long sequences due to their quadratic complexity in time and memory [11], which restricts their suitability for real-time tasks like remote robotic surgery. As a result, modifications of the Transformer framework, including Temporal Convolutional Networks (TCN) and Convolutional Self-Attention Networks, have been developed to manage long-sequence data with lower computational costs and enhanced prediction performance [20]. The Informer framework enhances performance by employing a ProbSparse self-attention mechanism, reducing computational complexity from $O(n^{2})$ to O(nlogn). This development in the Informer model makes it effective for handling long-sequence data in real-time settings [11]. Furthermore, the Informer’s generative-style decoder mitigates error accumulation in long-term predictions, enhancing its reliability for tracking positional changes over time [21]. These features make the Informer a robust choice for predicting the arm’s position of the PSM in remote robotic surgery. In such scenarios, challenges such as packet loss and jitter can disrupt performance, but the Informer’s design ensures stability and precision in PSM position estimation, even under challenging network conditions.

In addition to deep learning architectures, several recent studies have addressed broader network-related challenges encountered in telesurgery and remote robotic control systems. In [22], a review categorizes current strategies for addressing delay-related issues in telesurgery and telementoring into three key areas: network resource optimization, processing delay minimization, and delay-robust compensation techniques. Techniques such as federated learning, predictive control models, and VR-integrated compensation systems have been highlighted for their potential to reduce transmission latency, improve responsiveness at the edge, and mitigate feedback delays. A forecast-based recovery mechanism was proposed in [23] to improve real-time control during packet loss and delays. This method uses time-series prediction models such as Vector Auto-Regression (VAR) and Sequence-to-Sequence (Seq2Seq) learning to recover dropped command signals. Evaluated through simulations and real robotic experiments, the approach demonstrated improved trajectory tracking under burst loss conditions, addressing challenges such as command recovery, network delay tolerance, and stable control over unreliable links.

In [24], latency is identified as a critical factor in telesurgical precision, with 200 ms noted as a threshold beyond which safety and control degrade. Strategies for latency reduction include integrating 6G wireless technologies, quantum computing, edge processing, and enhancements in feedback handling. The study also emphasizes cybersecurity, legal, and operational challenges that must be addressed to support the large-scale adoption of telesurgery platforms.

To better contextualize the performance and architectural characteristics of the models discussed, Table 1 presents a comparative summary of RNN, LSTM, TCN, and the Informer model. The Informer stands out due to its ability to process long sequences efficiently, making it particularly well-suited for time-sensitive applications like remote robotic surgery.

In addition to these recent advances, the IEEE 1918.1 standard [2] outlines strict latency requirements for Tactile Internet applications, emphasizing that telesurgery falls within the medium-dynamic interaction category. For such scenarios, the acceptable end-to-end latency for haptic data exchange is within 10–100 ms. These latency thresholds are crucial for ensuring responsive and stable remote surgical interactions, underscoring the necessity of predictive and compensatory mechanisms in telesurgical systems operating over real networks.

## 3. Problem Statement

Remote robotic surgery is a promising application within the Tactile Internet. For remote surgical procedures to succeed, precise and real-time control of the PSM is essential. The PSM needs to accurately carry out commands from the SSM, including details such as position, orientation, linear and angular velocity, and the gripper angle of the surgical tools. However, sending the surgeon’s commands over a network can face several challenges, including packet loss, jitter, and delay. These issues, whether they happen as bursts or randomly, can seriously impact the reliability and accuracy of the PSM’s movements.

### 3.1. System Model

The proposed system model for remote robotic surgery integrates three key domains, as shown in Figure 1: the surgeon-side, patient-side, and network domains. Each domain plays a crucial role in the surgical workflow. The surgeon-side domain includes the surgeon console, which captures surgeon gestures and converts them into haptic commands. These commands represent the intended surgical movements, including force, orientation, and kinematic details. The commands are transmitted to the patient’s domain, where the PSM executes them. The PSM is equipped with a deep learning model (Informer, in our case) that estimates and corrects the robot’s arm position in real time. This ensures the surgeon’s movements are accurately replicated despite network-induced losses and delays. The network domain connects these two domains, providing reliable and low-latency communication necessary for the seamless execution of the surgery. This system enables precise remote surgeries, addressing distance and network variability challenges using TI technology.

### 3.2. System Overview

Let x(t) represent the state vector of the PSM at time *t*, which comprises several key operational parameters.(1)$$x\left(t\right)=\left[\begin{array}{c} p(t)\\ R(t)\\ v(t)\\ \omega (t)\\ \gamma (t) \end{array}\right]$$
where $p\left(t\right)={\left[x\left(t\right),y\left(t\right),z\left(t\right)\right]}^{T}$ is the 3D position vector of the PSM tool tip. $R\left(t\right)\in R^{3\times 3}$ is the orientation of the PSM, represented as a rotation matrix. $v\left(t\right)={\left[v_{x}\left(t\right),v_{y}\left(t\right),v_{z}\left(t\right)\right]}^{T}$ is the linear velocity vector. $\omega \left(t\right)={\left[\omega_{x}\left(t\right),\omega_{y}\left(t\right),\omega_{z}\left(t\right)\right]}^{T}$ is the angular velocity vector. γ(t) is the gripper angle of the PSM. The key parameter of interest in this study is the 3D position of the PSM’s robotic arm, represented as $p\left(t\right)={\left[x\left(t\right),y\left(t\right),z\left(t\right)\right]}^{T}$, where *t* denotes time. The goal is to ensure that p(t), as commanded by the SSM, is accurately executed by the PSM in real time. However, due to network imperfections, the state information p(t) received by the PSM can be incomplete or delayed, necessitating the need for a robust prediction model.

### 3.3. Network Errors and Challenges

In this paper, the focus is on addressing the critical network-induced errors that impact the performance of the PSM in Tactile Internet-enabled robotic surgery. Specifically, the following types of errors are considered.

Burst Errors: These errors occur in clusters, where multiple consecutive packets are lost, leading to significant gaps in transmitted data.Random Errors: These errors result in the loss of individual packets at random intervals, creating sporadic gaps in the transmitted position data.

Both errors can significantly degrade the PSM’s ability to replicate the SSM’s commands in real-time accurately. In our previous work [25,26], random errors were addressed using the Kalman filter (KF) for position estimation, which demonstrated effective compensation. However, the KF struggled to handle burst errors, highlighting the need for a more robust approach. In this study, the Informer predictive model is used to mitigate the impact of burst errors and enhance position estimation accuracy under these challenging conditions.

### 3.4. Optimization Problem

The position estimation task is formulated as an optimization problem that aims to improve the accuracy and performance of the PSM. The problem involves multiple parameters, including the tool tip’s 3D position (x,y,z), linear velocity, angular velocity, orientation, and gripper angle. However, the primary focus is on minimizing the state estimation error of the 3D position. Considering the complexity introduced by network-induced uncertainties, such as packet loss and jitter, this targeted approach ensures smooth, energy-efficient, and reliable operation under challenging network conditions. The optimization problem is expressed as follows:(2a)$$\begin{array}{cc} \; & \min_{\hat{p}\left(t\right)}\;E\left[\sum_{t=1}^{T}\left(∥\hat{p}{\left(t\right)-p\left(t\right)∥}^{2}+\alpha E\left(t\right)+\beta \Phi \left(t\right)\right)\right] \end{array}$$(2b)$$\begin{array}{cc} \; & s.t.\;∥\hat{p}\left(t\right)-\hat{p}\left(t-\Delta t\right)∥\leq \epsilon_{\mathrm{sync}},\;\forall t \end{array}$$(2c)$$\begin{array}{cc} \; & \hat{p}\left(t\right)\in P,\;p_{\min}\leq \hat{p}\left(t\right)\leq p_{\max},\;\forall t \end{array}$$(2d)$$\begin{array}{cc} \; & ∥v\left(t\right)∥\leq v_{\max},\;∥\omega \left(t\right)∥\leq \omega_{\max},\;\forall t \end{array}$$(2e)$$\begin{array}{cc} \; & ∥a\left(t\right)∥\leq a_{\max},\;a\left(t\right)=\frac{v(t)-v(t-1)}{\Delta t},\;\forall t \end{array}$$(2f)$$\begin{array}{cc} \; & \gamma_{\min}\leq \hat{\gamma }\left(t\right)\leq \gamma_{\max},\;\forall t \end{array}$$(2g)$$\begin{array}{cc} \; & \sum_{t=1}^{T}\left(\lambda_{v}{∥v\left(t\right)∥}^{2}+\lambda_{\omega }{∥\omega \left(t\right)∥}^{2}\right)\leq E_{\max}. \end{array}$$

The objective function in ([Disp-formula FD2a-sensors-25-03067]) minimizes the expected weighted sum of three components: state estimation error, energy consumption, and a penalty term for network-induced uncertainties. The state estimation error, represented as $∥\hat{p}{\left(t\right)-p\left(t\right)∥}^{2}$, ensures that the predicted position $\hat{p}\left(t\right)$ is as close as possible to the true position p(t) at each time step *t*. The energy consumption E(t), expressed as $\lambda_{v}{∥v\left(t\right)∥}^{2}+\lambda_{\omega }{∥\omega \left(t\right)∥}^{2}$, accounts for the contributions of linear velocity ∥v(t)∥ and angular velocity ∥ω(t)∥ to the total power usage. Here, $\lambda_{v}$ and $\lambda_{\omega }$ are weighting coefficients that balance the relative importance of these velocities. The penalty term Φ(t) quantifies the impact of network-induced uncertainties, such as packet loss and jitter, and is modeled using a four-state HMM. The trade-offs among accuracy, energy efficiency, and robustness are controlled by the weighting coefficients α and β.

The optimization problem is subject to several constraints to ensure operational feasibility. The real-time synchronization constraint in (2b) enforces that the variation between consecutive predicted positions is below a predefined threshold $\epsilon_{\mathrm{sync}}$, ensuring that the system operates in real-time. The position feasibility constraint in (2c) ensures that the predicted position $\hat{p}\left(t\right)$ lies within the predefined workspace P, bounded by $p_{\min}$ and $p_{\max}$.

The velocity limits in (2d) restrict the linear velocity ∥v(t)∥ and angular velocity ∥ω(t)∥ to their respective maximum allowable values $v_{\max}$ and $\omega_{\max}$. To maintain smooth transitions in position, the smoothness constraint in (2e) limits the acceleration a(t), calculated as $a\left(t\right)=\frac{v(t)-v(t-1)}{\Delta t}$, to an upper bound $a_{\max}$. The gripper angle constraint in (2f) ensures that the gripper angle $\hat{\gamma }\left(t\right)$ remains within operational limits $\left[\gamma_{\min},\gamma_{\max}\right]$. Finally, the energy budget constraint in (2g) ensures that the total energy consumption over the time horizon *T* does not exceed $E_{\max}$.

This optimization problem integrates multiple objectives and constraints to accurately estimate positions while maintaining smooth, energy-efficient, and reliable operations under network-induced uncertainties. However, several challenges arise in this context, including ensuring real-time execution, handling both random and burst packet loss effectively, and maintaining computational efficiency. These challenges are critical to achieving precise, smooth, and reliable remote robotic operations.

To address these challenges, we propose using the Transformer-based Informer predictive model. This model is designed to handle complex dependencies and real-time constraints efficiently, providing effective solutions for position estimation. The proposed approach ensures robustness and adaptability under challenging network conditions, improving the overall accuracy and performance of the PSM.

## 4. Proposed Model

This section discusses the proposed approach for solving the problem outlined above. The section is divided into two parts. Part A focuses on modeling network-induced packet loss using a four-state HMM to simulate realistic loss scenarios, and Part B explains the Informer framework, detailing its capabilities and structure.

### 4.1. Modeling Packet Loss

The different types of packet loss were defined in [27], and we have simulated packet loss in the network using the four-state HMM, where each state represents a specific network condition, such as:State 1 ($S_{1}$): Successful packet reception during a gap period.State 2 ($S_{2}$): Successful packet reception during a burst period.State 3 ($S_{3}$): Packet loss during a burst period.State 4 ($S_{4}$): Packet loss during a gap period.

Two probabilities govern state transitions.

Burst Density ($P_{B}$): Probability of entering or remaining in a burst state.Gap Density ($P_{G}$): Probability of entering or remaining in a gap state.

The following matrix T represents the transition probabilities.(3)$$T=\left[\begin{array}{cccc} 1-P_{B} & P_{B} & 0 & 0\\ 0 & 1-P_{G} & P_{G} & 0\\ 0 & 0 & 1-P_{G} & P_{G}\\ P_{B} & 0 & 0 & 1-P_{B} \end{array}\right]$$

The four-state HMM is selected for its ability to effectively model complex patterns of burst and random errors, providing a more detailed representation of packet loss dynamics. While simpler models like the Gilbert or two-State HMM exist [28,29], the four-state HMM offers a structured approach to distinguishing between packet loss and successful reception during burst and gap periods. This enables a more accurate simulation of network conditions, which is crucial for high-performance TI environments. Figure 2 provides a visual representation of the HMM state transitions and their integration into the Informer prediction framework.

To address the impact of network-induced uncertainties on position estimation, we first simulate packet loss in haptic data. A sequence of haptic data points representing the position of the SSM over time is defined as follows.(4)p(t)=[x(t),y(t),z(t)]
where t=1,2,…,T, and *T* is the sequence length. Packet loss is simulated by modifying this sequence based on the HMM state at each time step.

For States $S_{3}$ or $S_{4}$ (packet loss), the data point is set to zero.(5)$$\hat{p}\left(t\right)=\left[0,0,0\right]$$For States $S_{1}$ or $S_{2}$ (packet reception), the data point is preserved.(6)$$\hat{p}\left(t\right)=p\left(t\right)$$

The resulting sequence $\hat{p}\left(t\right)$, which contains missing values due to simulated packet loss, is then fed into the Informer model. The Informer model processes this sequence to predict the missing values and reconstruct an accurate estimate of the PSM’s position. This enables robust position estimation despite network-induced data loss, ensuring the system can effectively handle real-time uncertainties in haptic communication.

### 4.2. Informer Model-Based Predictive Approach

This study employs the Informer framework, a modified Transformer-based approach, to improve the real-time accuracy of PSM position predictions during remote robotic surgery, as illustrated in Figure 3. Traditional Transformers often struggle with handling long sequences due to their high computational demands and memory usage. The Informer addresses these limitations through key innovations, such as ProbSparse attention, self-attention distilling, and a generative-style decoder. While the Informer model was introduced in [11,21], we have applied our own optimization techniques to adapt it for our specific application. Below, we discuss the Informer model, and in the next section, we present how we implemented our techniques to enhance its performance.

### 4.3. Description of the Informer Model

This subsection provides a detailed description of each component of the Informer Model and its role in achieving efficient and accurate predictions.

#### 4.3.1. Optimized Attention Mechanism

In conventional self-attention, as introduced in [17], scaled dot-products are computed for queries, keys, and values.(7)$$A\left(Q,K,V\right)=\mathrm{Softmax}\left(\frac{QK^{T}}{\sqrt{d}}\right)V$$
where $Q\in R^{L_{Q}\times d}$, $K\in R^{L_{K}\times d}$, and $V\in R^{L_{V}\times d}$. For each query $q_{i}$, the attention mechanism is defined as.(8)$$A\left(q_{i},K,V\right)=\sum_{j}\frac{\exp\left(\frac{q_{i}k_{j}^{T}}{\sqrt{d}}\right)}{\sum_{l}\exp\left(\frac{q_{i}k_{l}^{T}}{\sqrt{d}}\right)}v_{j}$$

This method has a computational complexity of $O(L^{2})$, making it inefficient for lengthy sequences. To address this, the Informer introduces ProbSparse attention, which reduces computational requirements while retaining accuracy.

#### 4.3.2. Identifying Relevant Queries

To efficiently process extended sequences, the Informer uses a query sparsity measurement based on Kullback–Leibler (KL) divergence. For a given query $q_{i}$, the attention distribution $p(k_{j}|q_{i})$ is compared with a uniform distribution $u\left(k_{j}|q_{i}\right)=\frac{1}{L_{K}}$. The sparsity metric is defined as:(9)$$M\left(q_{i},K\right)=\ln\left(\sum_{j=1}^{L_{K}}\exp\left(\frac{q_{i}k_{j}^{T}}{\sqrt{d}}\right)\right)-\frac{1}{L_{K}}\sum_{j=1}^{L_{K}}\frac{q_{i}k_{j}^{T}}{\sqrt{d}}$$

This metric identifies the top queries that carry the most critical information.

#### 4.3.3. ProbSparse Attention Mechanism

Using the sparsity metric, the ProbSparse attention mechanism focuses only on significant queries. The updated attention mechanism is defined as:(10)$$A\left(Q,K,V\right)=\mathrm{Softmax}\left(\frac{QK^{T}}{\sqrt{d}}\right)V$$

Here, Q is a sparse matrix containing the top *u* queries selected based on M(q,K). The number of key queries *u* is determined using a factor *c*, set as $u=c\cdot \ln L_{Q}$, reducing the complexity to O(LlnL).

#### 4.3.4. Streamlined Self-Attention Distilling

The Informer applies self-attention distilling to simplify data processing. Input sequences are condensed layer by layer to emphasize key features and eliminate unnecessary information. The distilling process is described as:(11)$$X_{j+1}^{t}=\mathrm{MaxPool}\left(\mathrm{ELU}(\mathrm{Conv}1d\left(\left[X_{j}^{t}\right]AB\right))\right)$$
where $[X_{j}^{t}]AB$ represents the attention block, and Conv1d is a one-dimensional convolutional layer. This reduces memory usage to O((2−ϵ)LlnL).

#### 4.3.5. Efficient Encoder for Long Sequences

The Informer encoder efficiently processes long sequential inputs, balancing memory use and computational efficiency. Input sequences Xt are transformed into matrices $Xt^{\mathrm{en}}\in R^{L_{x}\times d\mathrm{model}}$, with self-attention distilling ensuring only essential features are retained. Inspired by techniques in dilated convolutions [30,31], the transformation from layer *j* to j+1 follows.(12)$$X{j+1}^{t}=\mathrm{MaxPool}\left(\mathrm{ELU}(\mathrm{Conv}1d\left({\left[X_{j}^{t}\right]}_{AB}\right))\right)$$

#### 4.3.6. Fast Generative Decoder

The decoder in the Informer model predicts entire sequences in a single pass, ultimately improving speed and reducing cumulative errors. Known tokens $X_{\mathrm{token}}$ and placeholders $X_{0}$ are used, with masked multi-head attention ensuring predictions remain causal:(13)$$X_{\mathrm{de}}^{t}=\mathrm{Concat}\left(X_{\mathrm{token}}^{t},X_{0}^{t}\right)$$

This design enables fast, precise predictions, making the Informer ideal for real-time applications like remote robotic surgery. The Informer model and its components are explained. In the next section, the optimization problem is integrated into the Informer framework to address the challenges in our work.

## 5. Integration of Optimization Problem

Building on the optimization problem formulated in Section 3, Part B, it is now integrated with the Transformer-based Informer model. The Informer model serves as the foundation for solving this problem by directly embedding the constraints and objectives into its training process. This approach aligns with treating optimization as a differentiable layer, as introduced in [32], and is further guided by insights from optimization principles reviewed in [33]. By combining the strengths of the Informer model and optimization techniques, this approach enhances predictive accuracy.

### 5.1. Optimization as a Layer

The optimization problem, defined to minimize the state estimation error $∥\hat{p}{\left(t\right)-p\left(t\right)∥}^{2}$, while satisfying constraints like energy efficiency and smoothness, is modeled as a differentiable layer. This layer translates as follows.

Objective Function: Position accuracy and energy efficiency are incorporated as primary and secondary terms in the loss function:(14)$$L_{\mathrm{total}}=L_{\mathrm{pos}}+\alpha E\left(t\right)+\gamma {∥a\left(t\right)∥}^{2}+\beta \Phi \left(t\right)$$
where E(t) represents energy consumption, ∥a(t)∥ ensures smoothness, and Φ(t) addresses robustness to network uncertainties.Constraints: Operational feasibility is maintained through penalties for violating constraints, ensuring the model adheres to real-time requirements.

### 5.2. ProbSparse Attention for Position-Critical Features

The ProbSparse attention mechanism in the Informer framework is designed to focus computational resources on the most relevant input features, enabling efficient processing of complex data dependencies. To align the attention mechanism with the optimization problem, the sparsity metric is modified to prioritize position-critical features while deemphasizing secondary features, such as velocity (v(t)) or angular velocity (ω(t)). The updated sparsity metric is defined as.(15)$$M_{\mathrm{pos}}\left(q_{i},K\right)=M\left(q_{i},K\right)+\lambda_{1}e_{x}{\left(t\right)}^{T}We_{x}\left(t\right)$$
where $M(q_{i},K)$ is the original sparsity metric for attention weights; $e_{x}\left(t\right)=p\left(t\right)-\hat{p}\left(t\right)$ is the state estimation error for the tooltip position; $\lambda_{1}$ is a scaling factor that prioritizes position estimation error; and *W* is the weighting matrix to emphasize certain components of x,y,z. This modification ensures that the attention mechanism emphasizes features that reduce the position estimation error directly. The original ProbSparse attention mechanism has a complexity of O(LlogL), where *L* is the sequence length. The modification to the sparsity metric adds the position-related term $e_{x}{\left(t\right)}^{T}We_{x}\left(t\right)$, which involves a matrix-vector multiplication. Since this operation has constant time complexity concerning *L*, the overall complexity of the attention mechanism remains unchanged at O(LlogL). This ensures that the attention mechanism remains efficient while prioritizing position-critical features.

### 5.3. Encoder-Guided Constraint Adherence

The encoder processes input sequences, including corrupted or incomplete position data $\hat{p}\left(t\right)$ arising from packet loss modeled by a four-state HMM. To ensure that the latent representations align with the constraints of the optimization problem, the encoder incorporates smoothness and energy constraints as regularization terms. The encoder loss function is designed to minimize.(16)$$L_{\mathrm{enc}}=E\left[∥{\hat{X}}_{\mathrm{enc}}-X_{\mathrm{true}}{∥}^{2}+\gamma_{1}{∥a\left(t\right)∥}^{2}\right]$$
where ${\hat{X}}_{\mathrm{enc}}$ is the latent representation generated by the encoder. $X_{\mathrm{true}}$ is the ground truth latent representation. a(t)=(v(t)−v(t−1))/Δt is the acceleration, penalized to enforce smooth movements. $\gamma_{1}$ is a regularization weight controlling smoothness constraints.

This approach ensures that the encoder generates feasible latent representations. Moreover, incorporating regularization terms, such as the acceleration penalty ${∥a\left(t\right)∥}^{2}={∥v\left(t\right)-v\left(t-1\right)∥}^{2}/\Delta t^{2}$, introduces only constant-time computations per timestep *t*. These additional operations do not depend on the sequence length *L* and, therefore, have a negligible impact on complexity.

### 5.4. Decoder for Real-Time Position Estimation

The Informer decoder reconstructs the estimated position sequence $\left(\hat{x}\left(t\right),\hat{y}\left(t\right),\hat{z}\left(t\right)\right)$ by aligning its predictions with the optimization objectives. The decoder’s loss function incorporates the state estimation error as the primary term and energy consumption as a secondary constraint.(17)$$L_{\mathrm{dec}}=\frac{1}{T}\sum_{t=1}^{T}{∥\hat{p}\left(t\right)-p_{\mathrm{true}}\left(t\right)∥}^{2}+\delta_{1}E\left(t\right)$$
where $p_{\mathrm{true}}\left(t\right)$ is the true 3D position of the tool tip at time *t*; $E\left(t\right)=\lambda_{v}{∥v\left(t\right)∥}^{2}+\lambda_{\omega }{∥\omega \left(t\right)∥}^{2}$ represents the energy consumption at time *t*; and $\delta_{1}$ is a penalty term for energy efficiency constraints. This formulation ensures that the predicted positions minimize estimation error while adhering to energy constraints, enabling real-time execution. The decoder reconstructs position sequences with a complexity of O(LlogL), driven by masked self-attention and feedforward operations. Adding energy consumption terms $E\left(t\right)=\lambda_{v}{∥v\left(t\right)∥}^{2}+\lambda_{\omega }{∥\omega \left(t\right)∥}^{2}$ in the loss function introduces constant-time operations per timestep *t*. Since these computations are independent of the sequence length *L*, the decoder’s complexity remains unchanged.

### 5.5. Incorporating Network Information

The problem in Section 4 can be further enhanced by explicitly incorporating network parameters such as latency, jitter, and packet loss to better align with the requirements. These factors directly impact the real-time performance and robustness of position estimation in the PSM. To address this, the robustness term Φ(t) is redefined to include these network-specific metrics.(18)$$\Phi \left(t\right)\;=[\eta_{1}\;\mathrm{PacketLossRate}\left(t\right)\;+\;\eta_{2}\mathrm{Latency}\left(t\right)\;+\;\eta_{3}\mathrm{Jitter}\left(t\right)]$$
where PacketLossRate(t) is a proportion of packets lost at time *t*; Latency(t) is an end-to-end delay of packets at time *t*; Jitter(t) is the variability in packet inter-arrival times at time *t*; and $\eta_{1},\eta_{2},\eta_{3}$ are weights controlling the contribution of each network parameter to the robustness term. Incorporating network features as an auxiliary input increases the dimensionality of the input data but does not affect the sequence length *L*. The complexity of the Informer remains proportional to O(LlogL), as the primary cost arises from processing the sequence length, not the dimensionality of the input. Therefore, adding network features has a negligible impact on the overall complexity. The Informer model is augmented to process network information alongside position data to improve its robustness and adaptability. Network parameters such as predicted latency and jitter are included as auxiliary inputs to the model.(19)$$\;\;\hat{p}\left(t\right)\;=\;f_{\mathrm{Informer}}\left(X_{\mathrm{input}},\mathrm{PredictedLatency},\mathrm{PredictedJitter}\right)$$
where $X_{\mathrm{input}}$ is the input sequence containing the corrupted or incomplete position data. PredictedLatency, PredictedJitter are predicted network conditions at time *t*, which are included as additional features. The encoder is modified to account for both position and network conditions. The encoder loss function becomes:(20)$$L_{\mathrm{enc}}=E\left[∥{\hat{X}}_{\mathrm{enc}}-X_{\mathrm{true}}{∥}^{2}+\gamma_{1}{∥a\left(t\right)∥}^{2}+\gamma_{2}\mathrm{LatencyPenalty}\right]$$
where $\gamma_{2}$ weights the penalty for high latency, ensuring the model learns to operate efficiently under varying network conditions. Network conditions such as latency, jitter, and packet loss are simulated using a four-state HMM to evaluate the Informer’s performance under realistic TI scenarios. These simulations help generate realistic data for training and testing the model. The robustness term Φ(t) and the augmented input features allow the Informer to adapt its predictions dynamically.

### 5.6. Solvability and Convergence Analysis

This section presents a theoretical analysis of the solvability and convergence of the optimization-enhanced Informer model. The analysis is based on the optimization problem formulated in Section 3.4:(21)$$\min_{\hat{p}\left(t\right)}E\left[\sum_{t=1}^{T}\left(∥\hat{p}{\left(t\right)-p\left(t\right)∥}^{2}+\alpha E\left(t\right)+\beta \Phi \left(t\right)\right)\right]$$
subject to a set of convex constraints $C=\{c_{i}\left(\hat{p}\left(t\right),v\left(t\right),\omega \left(t\right),\gamma \left(t\right)\right)\leq 0\}$.

**Theorem** **1**(Existence and Uniqueness). *If the feasible region F defined by C is convex and the objective function is strictly convex, then a unique global minimizer ${\hat{p}}^{*}\left(t\right)$ exists.*

**Proof.** The objective includes a squared Euclidean norm $∥\hat{p}{\left(t\right)-p\left(t\right)∥}^{2}$, which is strictly convex. The energy E(t) and robustness term Φ(t) are quadratic or linear, hence convex. Since convex functions over a convex domain ensure a unique minimizer, the problem is solvable.    □

The optimization-enhanced loss function used in Informer training is:(22)$$L_{\mathrm{total}}=\sum_{t=1}^{T}\left(∥\hat{p}{\left(t\right)-p\left(t\right)∥}^{2}+\alpha E\left(t\right)+\gamma {∥a\left(t\right)∥}^{2}+\beta \Phi \left(t\right)\right)$$

**Theorem** **2**(Gradient Descent Convergence). *Assuming $L_{\mathrm{total}}$ is Lipschitz-smooth and convex concerning the model parameters, gradient descent with learning rate $\eta <\frac{2}{L}$ converges to the global minimum.*

**Proof.** Let *L* be the Lipschitz constant of the gradient $\nabla L_{\mathrm{total}}$. The standard convergence result for convex, smooth functions ensures that gradient descent updates of the form:(23)$$\theta^{\left(k+1\right)}=\theta^{\left(k\right)}-\eta \nabla L_{\mathrm{total}}\left(\theta^{\left(k\right)}\right)$$Note (Equation 23) monotonically decreases the loss if $\eta \in (0,\frac{2}{L})$, and converges to a global minimum due to convexity.    □

## 6. Experimental Setup and Results

### 6.1. Dataset

The JIGSAWS dataset [12] is a publicly available resource containing data from surgical tasks performed using the da Vinci robotic surgical system. This dataset includes synchronized kinematic data, video recordings, and gesture annotations. It was collected during three core surgical tasks, such as knot-tying, suturing, and needle-passing, performed on a bench-top model by eight surgeons with varying skill levels categorized as expert, intermediate, and novice. For this study, 39 trials of the knot-tying task were selected for evaluation. The kinematic data in the JIGSAWS dataset provides Cartesian positions ($p\in R^{3}$), rotation matrices ($R\in R^{3\times 3}$), linear velocities ($v\in R^{3}$), rotational velocities ($\omega \in R^{3}$), and grasper angles (θ) for both the left and right tools. These features correspond to the PSM and SSM, resulting in 76 attributes sampled at 30 Hz.

### 6.2. Simulation Setup

The simulations were conducted on a system equipped with an Intel Core i7 processor and 32 GB of RAM, running Ubuntu 22.04 LTS (Linux OS). The experiments were implemented in Python 3.10, using PyTorch 2.0.1 as the primary deep learning framework.

The experiments were carried out in Jupyter Notebook 6.5.4, which was used for coding and execution. Data preprocessing, model training, and evaluations were performed entirely within this environment.

The overall pipeline of our Python-based implementation, from dataset preparation to model evaluation, is presented in Algorithm 1.
**Algorithm 1** Informer-Based Position Prediction under HMM-Induced Packet Loss1:**Input:** Kinematic data, number of time steps *T*, HMM parameters $P_{B}$, $P_{G}$2:**Output:** Predicted position $\hat{p}\left(t\right)$, performance metrics (MAE, MSE, RMSE)3:**Initialization:**4:Load Cartesian position data p(t) from JIGSAWS dataset5:Define 4-state HMM using $P_{B}$, $P_{G}$ transition probabilities6:Simulate packet loss to create corrupted data $\tilde{p}\left(t\right)$7:**for** t=1 to *T* **do**8:      **if** HMM state at *t* is S3 or S4 **then**9:          $\tilde{p}\left(t\right)\leftarrow \left[0,0,0\right]$10:      **else**11:          $\tilde{p}\left(t\right)\leftarrow p\left(t\right)$12:      **end if**13:**end for**14:Normalize and preprocess $\tilde{p}\left(t\right)$15:Split into training and testing sets16:Initialize Informer model using PyTorch17:Train Informer: $\tilde{p}\left(t\right)$ as input, p(t) as target18:Predict: $\hat{p}\left(t\right)\leftarrow$ Informer($\tilde{p}\left(t\right)$)19:**Evaluation:**20:Compute MAE, MSE, RMSE for *x*, *y*, and *z*21:Compare with LSTM, RNN, TCN models

To replicate the effects of unstable network conditions encountered in TI-based surgery, packet loss patterns were applied using a four-state HMM. These patterns were mapped onto haptic position data, where lost packets were either removed or replaced with zero vectors. The Informer model was then trained to reconstruct the missing data, enabling robust position estimation even under network-induced disruptions.

### 6.3. Results and Discussion

#### 6.3.1. Impact of Packet Loss on Position Estimation

Part 1 of Figure 4 illustrates the packet loss pattern over 1000 time steps. The red spikes indicate the moments when packets were lost (with a value of 1) and successfully received (value of 0). This pattern highlights packet loss’s irregular and bursty nature, effectively simulating real-world network conditions.

Parts 2–4 of Figure 4 depict the original and corrupted positions of the robotic arm’s tool tip along the X, Y, and Z axes under simulated packet loss conditions. The solid black line represents the ground truth position, while the red dashed line illustrates the corrupted position data resulting from packet loss. The gray areas indicate the time intervals during which packet loss occurs.

#### 6.3.2. Performance of the Model Under Packet Loss

Figure 5 shows how the Informer model predicts the robotic arm’s tool tip position along all axes in bursty packet loss. Each plot compares the ground truth and the estimated position for 200 test time steps.

The X position prediction accuracy is 96.68%. The model achieves high accuracy in predicting the X-axis position. The predicted position closely follows the actual position, with very few deviations, indicating that the model handles packet loss well for this axis.Y position prediction accuracy is 95.96%. Similarly, the model performs effectively in predicting the Y-axis position. The predicted values align almost perfectly with the actual values, except for minor deviations during sharp transitions, demonstrating the robustness of the model.Z position prediction accuracy is 90.37%. The Z-axis shows a slightly lower accuracy than the X and Y axes, with some noticeable deviations during time steps where the actual position exhibits rapid changes. However, the overall prediction still captures the trend of position movements, showing that the model can still predict reasonably well in challenging packet loss scenarios.

The slightly lower Z-axis accuracy was observed specifically under burst packet loss conditions, which are more difficult for the model to recover from. This level of accuracy is still reasonable within simulation-based studies, especially considering the real-time nature of predictions. The knot-tying task also involves limited Z-motion, which results in fewer training cues for that direction. Additionally, the dataset was recorded using the dVRK, an early research platform, which may introduce more noise along the Z-axis. At this stage, no axis-specific improvements were applied. In future work, improvements such as data augmentation or axis-weighted training will be considered to enhance Z-direction performance. Although this work focuses on the effect of network-induced loss, it is acknowledged that real robotic systems also face other sources of uncertainty, such as mechanical lag or sensor noise. Future versions of this framework will consider these combined effects to better assess the model’s robustness in practical use.

In Table 2, the Informer model’s performance is evaluated under varying network conditions, including different burst densities, gap densities, burst lengths, and gap lengths. For comparison, the Informer framework was evaluated alongside other deep learning models using the same data subset, with the results summarized in Table 3. The Informer model outperforms TCN, RNN, and LSTM in predicting tool tip positions under packet loss scenarios. Its ProbSparse attention mechanism reduces the computational complexity from $O(L^{2})$ in traditional self-attention to O(LlogL). TCN, with a complexity of O(L·k) (where *k* is the filter size), struggles with fixed receptive fields, making it less effective in burst error scenarios. RNNs, with a complexity of $O(L\cdot d^{2})$, are hindered by vanishing gradients, which limit their ability to capture temporal correlations. LSTMs address this issue with gating mechanisms but have the same complexity $O(L\cdot d^{2})$ and higher computational costs due to sequential processing. These characteristics make the Informer the most effective model for this task.

All models were evaluated under the same dataset split and packet loss conditions. Moreover, they were trained and tested on the same subset of the JIGSAWS dataset, using separate kinematic files for training and testing. The same packet loss patterns, generated using a four-state HMM, were applied uniformly across all models. Identical evaluation settings and training conditions were used to ensure a fair comparison.

## 7. Conclusions

This paper presented a predictive approach using the Transformer-based Informer model to enhance position estimation accuracy in remote robotic surgery. A four-state HMM was employed to simulate realistic packet loss scenarios, addressing both burst and random loss conditions. The Informer model effectively mitigated network-induced uncertainties, such as jitter and delay, ensuring accurate real-time predictions. Experimental results demonstrated a prediction accuracy exceeding 90% under diverse network conditions, outperforming traditional models like LSTM and RNN. The integration of constraints such as energy efficiency, smoothness, and robustness further validated the model’s suitability for Tactile Internet-enabled surgical applications.

Future work will focus on several key areas. First, the framework can be extended to incorporate multi-objective optimization, balancing position estimation accuracy, latency reduction, and surgical task precision. Second, adaptive mechanisms will be explored to dynamically address varying network conditions, including fluctuating levels of jitter, delay, and packet loss. Lastly, the model’s generalizability will be evaluated through cross-domain validation on diverse surgical datasets and tasks beyond knot-tying, ensuring its applicability to a broader range of robotic-assisted medical procedures.Additionally, future work will include ablation studies to evaluate the contribution of different components of the proposed framework.

This framework offers the advantage of high prediction accuracy under burst packet loss while maintaining low computational complexity. However, its performance in the Z-axis is slightly lower. Although the study demonstrates strong performance in simulated environments, the value of real-world validation is fully recognized. In future work, the aim is to deploy the proposed framework on a physical robotic platform to assess its real-time effectiveness under actual network conditions. This step will help translate the predictive model from theory to practical impact in robotic-assisted surgical systems.

## Figures and Tables

**Figure 1 sensors-25-03067-f001:**
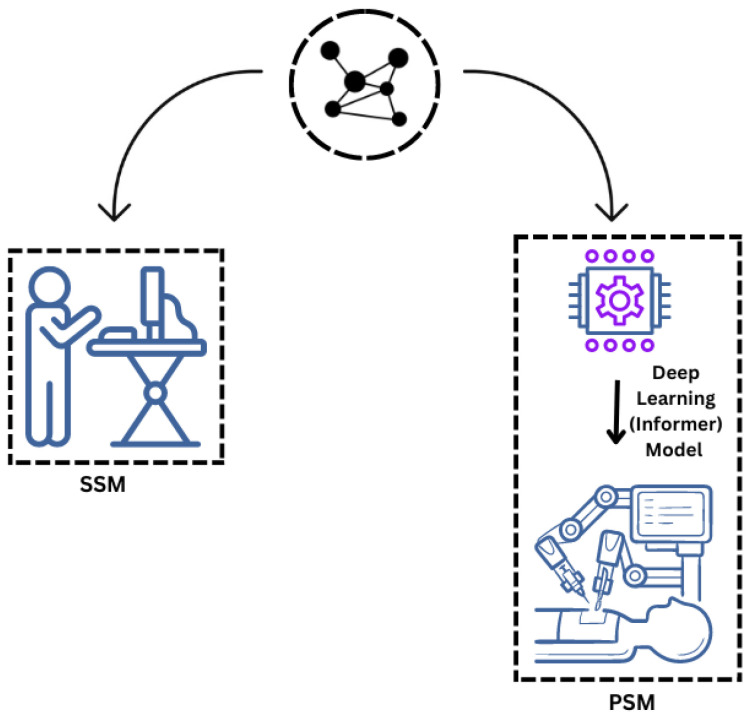
Remote robotic surgery framework utilizing TI and Informer model for enhanced PSM precision.

**Figure 2 sensors-25-03067-f002:**
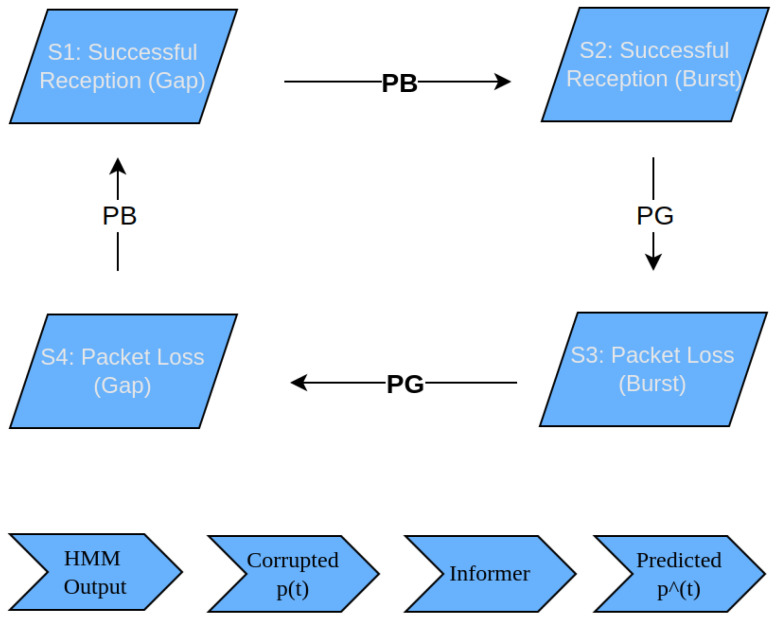
The 4-State HMM representing packet reception and loss during burst and gap periods. State transitions are governed by burst density ($P_{B}$) and gap density ($P_{G}$). The lower section illustrates how HMM output feeds into the Informer model for position prediction.

**Figure 3 sensors-25-03067-f003:**
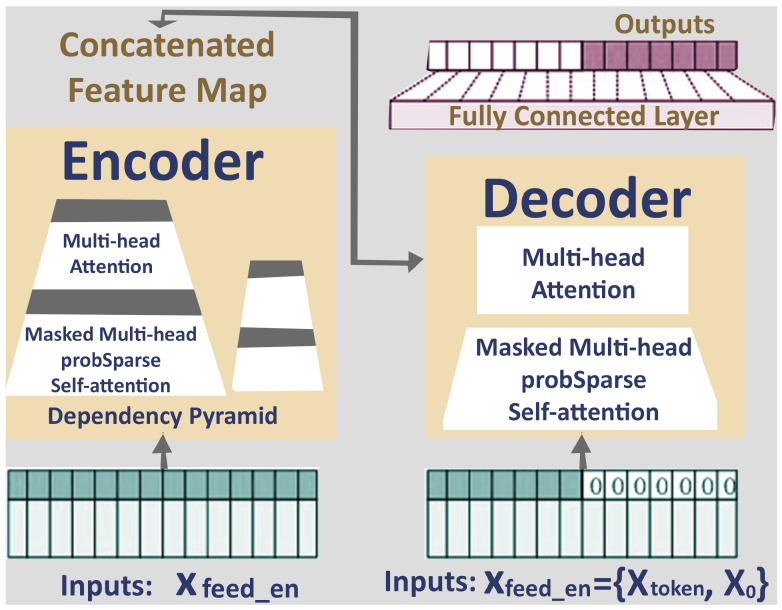
Informer model Encoder-Decoder framework with ProbSparse attention mechanism.

**Figure 4 sensors-25-03067-f004:**
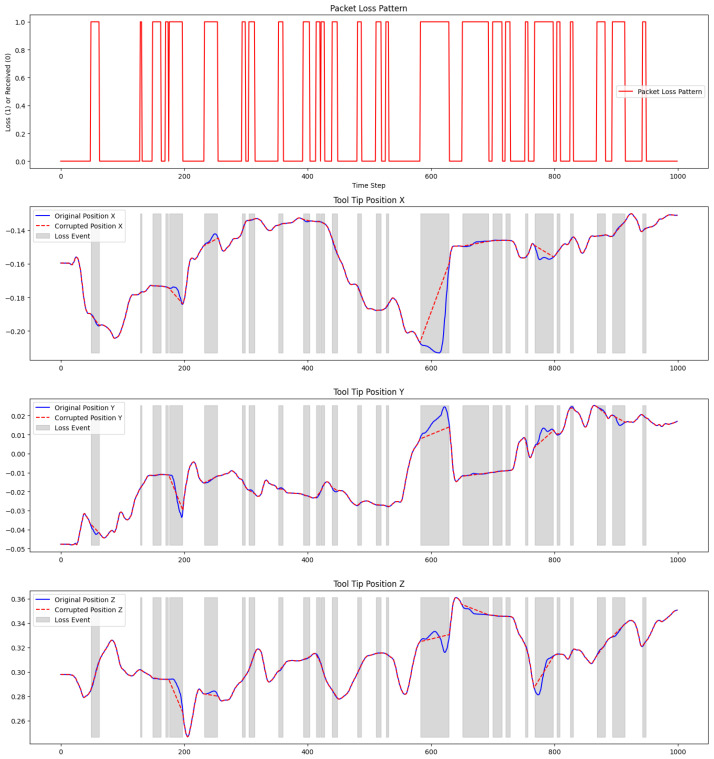
The top plot (Part 1) shows the simulated packet loss pattern across 1000 time steps. The following plots (Parts 2–4) display the original and corrupted tool tip position along all three axes, with gray-shaded regions highlighting periods of packet loss.

**Figure 5 sensors-25-03067-f005:**
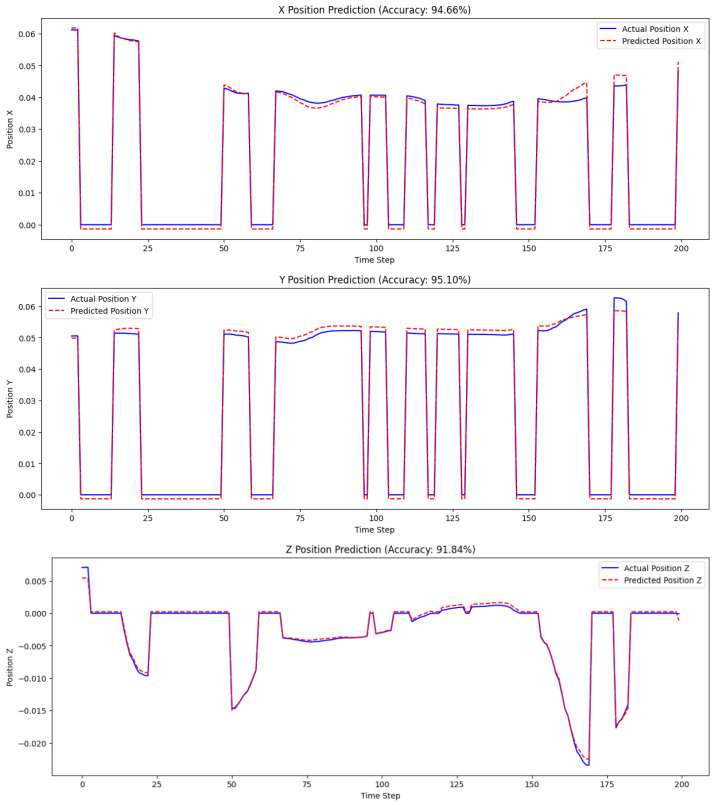
Prediction performance of the Informer model under packet loss for tool tip position in X, Y, and Z axes. Solid and dashed lines represent actual and predicted positions, respectively. The model achieves accuracies of 96.68%, 95.96%, and 90.37% for the X, Y, and Z axes, demonstrating robustness against network-induced packet loss.

**Table 1 sensors-25-03067-t001:** Comparison of deep learning models for sequence modeling.

Model	Type	Complexity	Sequential Handling	Suitability for Long Sequences	Strength
RNN	Recurrent	$O(n\cdot d^{2})$	Yes	Moderate	Simplicity
LSTM	Recurrent	$O(n\cdot d^{2})$	Yes	Good	Handles long-term dependencies
TCN	Convolutional	O(n·logn)	No	Good	Parallelizable with long-range capture via dilation
Informer	Transformer	O(nlogn)	No	Excellent	Handles long sequences with reduced cost

**Table 2 sensors-25-03067-t002:** Informer model performance metrics at varying burst and gap densities, burst lengths, and gap lengths.

Burst Density	Gap Density	Burst Length	Gap Length	MSE	MAE	RMSE	Accuracy X (%)	Accuracy Y (%)	Accuracy Z (%)
0.3	0.95	4	8	0.0105	0.0725	0.1027	94.27	94.25	93.40
0.4	0.90	5	7	0.0119	0.0771	0.1090	93.45	92.30	91.22
0.5	0.85	6	6	0.0116	0.0768	0.1078	92.88	91.78	90.33
0.6	0.80	8	5	0.0123	0.0785	0.1109	91.33	90.22	89.12
0.7	0.75	10	4	0.0130	0.0792	0.1131	90.50	89.45	88.55
0.8	0.70	12	3	0.0136	0.0811	0.1166	89.12	88.90	87.50

**Table 3 sensors-25-03067-t003:** Comparison of deep learning models for position estimation.

Model	MSE	MAE
Informer	0.0192	0.1082
TCN	0.0724	0.1313
RNN	0.1368	0.1982
LSTM	0.1472	0.2004

## Data Availability

We used a publicly available dataset, which can be found at https://cirl.lcsr.jhu.edu/research/hmm/datasets/jigsaws_release/ (accessed on 28 October 2024).

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
