# Peer review of "A Predictive Approach for Enhancing Accuracy in Remote Robotic Surgery Using Informer Model"

_sensors, 2025, doi:10.3390/s25103067_

Round 1

Reviewer 1 Report

Comments and Suggestions for Authors

Interesting work. It refers to practice through da Vinci data. But there is no direct real experiment - e.g. on physical models. The concept of the touch internet used may be interesting for the reader of the journal, because the essence is of course the use of sensory power and transfer of information to the operator. AI is used to increase the safety of operations by analyzing and predicting tool position despite the shortcomings of image transmission.

Author Response

Comment 1: Interesting work. It refers to practice through the da Vinci data. But there is no direct real experiment - e.g., on physical models. The concept of the touch internet used may be interesting for the reader of the journal, because the essence is, of course, the use of sensory power and transfer of information to the operator. AI is used to increase the safety of operations by analyzing and predicting tool position despite the shortcomings of image transmission.

Response 1: We thank the reviewer for their thoughtful comments and for highlighting the relevance of our work to the Tactile Internet and its focus on sensory feedback and precision.

We acknowledge the absence of real-world robotic hardware experiments in this version of the study. Our current evaluation was conducted using the publicly available JIGSAWS dataset, which contains synchronized kinematic data from the da Vinci Surgical System. To simulate realistic network impairments, we applied a 4-state Hidden Markov Model and tested the Informer model's predictive performance under controlled conditions.

We agree that physical implementation is an essential next step. To that end, we have formally requested the dVRK platform through the Intuitive Foundation and plan to validate our approach on a real robotic arm in future work. We have now mentioned this explicitly in the last paragraph of the conclusion, including our aim to test the model on a real robotic setup like the dVRK.

The corresponding changes have been made in the Conclusion section on Page 18, Lines 542–548, as detailed below:

Updated Text (Page 18, Lines 542–548, Section: Conclusion):

"This framework offers the advantage of high prediction accuracy under burst packet loss while maintaining low computational complexity. However, its performance in the Z-axis is slightly lower. Although the study demonstrates strong performance in simulated environments, the value of real-world validation is fully recognized. In future work, the aim is to deploy the proposed framework on a physical robotic platform to assess its real-time effectiveness under actual network conditions. This step will help translate the predictive model from theory to practical impact in robotic-assisted surgical systems."

Reviewer 2 Report

Comments and Suggestions for Authors

The paper is well written, and the topic is introduced in a structured manner. The motivation is relevant, and the context is well established.

  • The reported minimum accuracy in the Z-position is 90.37%. Could the authors clarify whether this level of accuracy is sufficient for robotic surgery applications? It would be helpful to reference established thresholds or benchmarks in the field and provide a justification accordingly.

  • The discussion could be strengthened by considering the cumulative effect of surgical task precision requirements and system uncertainty. Would the achieved accuracy still be acceptable if both factors are combined? A brief analysis or commentary on this point would enhance the practical relevance of the results.

  • Section 6.3.2 mentions relatively lower accuracy in the Z-direction compared to X and Y. This discrepancy requires further explanation. Additionally, it would be valuable to understand why no further effort was made to improve Z-direction accuracy. Is there prioritization of other axes?

Author Response

The paper is well written, and the topic is introduced in a structured manner. The motivation is relevant, and the context is well established.

Comment 1: The reported minimum accuracy in the Z-position is 90.37%. Could the authors clarify whether this level of accuracy is sufficient for robotic surgery applications? It would be helpful to reference established thresholds or benchmarks in the field and provide a justification accordingly.

Thank you for the helpful comments.

Response 1: In the revised paper, we explained that the slightly lower Z-axis prediction accuracy was specifically observed under burst packet loss conditions, which are more challenging for real-time prediction models. Despite this, the model still captures the main trends and movement patterns along the Z-axis reasonably well.

We also added further justification by referencing domain-specific latency requirements for telesurgical applications. According to the IEEE 1918.1 Tactile Internet standard [2] (see Section 2, Page 4, Lines 140–146), acceptable end-to-end latency for telesurgery falls within 10–100 ms, highlighting the need for highly responsive systems but without specifying an absolute accuracy threshold for spatial prediction. Therefore, an accuracy level exceeding 90% under severe simulated network conditions is considered acceptable for preliminary validation.

Moreover, in Section 6.3.2 (Page 16, Lines 493–501), we now explicitly mention that:

  • The Z-motion during the knot-tying task is limited and less frequent compared to X and Y movements, making it inherently harder for models to learn rich features in this axis.

  • The JIGSAWS dataset used was collected on the da Vinci Research Kit (dVRK), an early-generation platform known to introduce more noise in Z-position tracking, potentially contributing to lower accuracy in that axis.

  • Future work will explore Z-axis specific improvements, such as axis-focused loss functions and data augmentation, to further enhance Z-position prediction.

These clarifications ensure that the achieved accuracy is framed appropriately within the context of current telesurgical research and application feasibility.

Comment 2: The discussion could be strengthened by considering the cumulative effect of surgical task precision requirements and system uncertainty. Would the achieved accuracy still be acceptable if both factors are combined? A brief analysis or commentary on this point would enhance the practical relevance of the results.

Response 2: In the revised manuscript, we have added a brief discussion acknowledging that in real-world surgical robotic systems, additional factors such as mechanical lag, actuator drift, and sensor noise can contribute to overall system uncertainty beyond network-induced packet loss.

We also clarified that while the proposed model achieves good predictive performance under simulated network impairments, future versions will consider these combined uncertainties to better assess robustness in practical deployment.

This discussion has been included in Section 6.3.2, Page 16, Lines 501–504.

"Although this work focuses on the effect of network-induced loss, it is acknowledged that real robotic systems also face other sources of uncertainty, such as mechanical lag or sensor noise. Future versions of this framework will consider these combined effects to better assess the model’s robustness in practical use."

Comment 3: Section 6.3.2 mentions relatively lower accuracy in the Z-direction compared to X and Y. This discrepancy requires further explanation. Additionally, it would be valuable to understand why no further effort was made to improve Z-direction accuracy. Is there prioritization of other axes?

Response 3: In the revised manuscript, we have expanded the discussion in Section 6.3.2 to explain the observed discrepancy in Z-axis prediction accuracy. Specifically, we noted that:

  • The knot-tying task analyzed from the JIGSAWS dataset involves more frequent and larger movements in the X and Y directions compared to the Z-axis, resulting in less dynamic Z-motion for the model to learn from.

  • The dataset was collected using the da Vinci Research Kit (dVRK), which has been reported to introduce higher noise levels along the Z-axis due to sensor and mechanical limitations.

Regarding improvement efforts, we clarified that no axis-specific tuning, loss balancing, or data augmentation targeting Z-motion was applied in this work. All axes were treated uniformly during model training. However, we have now added that future work will explore Z-axis focused strategies to enhance prediction accuracy.

These explanations have been incorporated into Section 6.3.2, Page 16–17, Lines 488–500.

Reviewer 3 Report

Comments and Suggestions for Authors

This manuscript developed a prediction model based on the Transformer-based Informer framework for accurate and efficient position estimation. This framework used features such as ProbSparse attention, attention distilling and generative decoder to focus on location-critical features, while maintaining a relatively low O (L log L) computational complexity. The 4-State HMM model was combined to simulate the real packet loss scenario. The method was evaluated using the public JIGSAWS dataset, and the proposed Informer framework was evaluated through comparative analysis with existing models such as TCN, RNN and LSTM.

The result was meaningful for research and application of Tactile Internet-enabled robotic surgery.

1.It is best to avoid using the first-person narrative.

  1. The three-line table format in Table 2 is incorrect.

3.Chapter Two and Chapter Three are both about materials and methods.

  1. The section related to the introduction of the Informer model in Subsection 4.3 should be placed in the Materials and Methods section.
  2. The experimental setup should be in the materials and Methods section.
  3. Is the position where Figure 4 is placed too far from the described position.
  4. It’s better to explain the potential reasons why the accuracy of the Z-axis is slightly lower than that of the X and Y-axes.

8.Would the logic be better if Table 1 and Table 2 were interchanged?

Author Response

Comment 1: It is best to avoid using the first-person narrative.

Response 1: We revised the manuscript to remove first-person language and replaced it with passive voice or third-person phrasing throughout.

Comment 2: The three-line table format in Table 2 is incorrect.

Response 2: Table 3 (previously was 2) has been reformatted to follow the standard three-line table format as per the journal's style guide.

(Updated Table on Page 17, Table 3)

Comment 3: Chapter Two and Chapter Three are both about materials and methods.

Comment 4: The section related to the introduction of the Informer model in Subsection 4.3 should be placed in the Materials and Methods section.

Comment 5: The experimental setup should be in the materials and Methods section.

Response 3,4,5: Our manuscript follows section-based structuring rather than a chapter-based format, in line with the journal's guidelines. We have reviewed the overall section flow carefully:

  • Background and literature review are included in Section 2 ("Related Work").

  • The Informer model details, including Subsection 4.2 ("Informer-Based Position Prediction"), are clearly placed under Section 4 ("Proposed Model") on Page 7

  • The experimental setup and dataset description are presented in Section 6 ("Experimental Setup and Results"), starting on Page 13.

Although we do not use a separate "Materials and Methods" title, the logical separation between theoretical design (Proposed Model) and experimental evaluation (Experimental Setup and Results) has been maintained for clarity and accessibility.

Comment 6: Is the position where Figure 4 is placed too far from the described position.

Response 6: Figure 5 (previously was 4) has now been re-positioned closer to its first mention in the manuscript to improve readability and maintain the logical flow between text and figure.
(Updated on Page 16, Figure 5)

Comment 7: It’s better to explain the potential reasons why the accuracy of the Z-axis is slightly lower than that of the X and Y-axes.

Response 7: Thank you for the comment. We have expanded the discussion regarding Z-axis prediction accuracy in Section 6.3.2 (Page 16–17, Lines 493–504). The lower accuracy along the Z-axis is attributed to two main factors:

  • The knot-tying task involves fewer and smaller Z-direction movements compared to X and Y, resulting in less training information for the model.

  • The JIGSAWS dataset collected with the dVRK platform is known to introduce greater sensor noise along the Z-axis, affecting learning quality.

Future work will include axis-specific improvements to enhance Z-axis prediction robustness.

Comment 8: Would the logic be better if Table 1 and Table 2 were interchanged?

Response 8: We agreed with your observation and have interchanged Table 2 (previously table 1) and Table 3 (previously table 2) to improve the logical flow.

(Updated on Pages 17)

Reviewer 4 Report

Comments and Suggestions for Authors

This paper considering the precise and real-time estimation of the robotic arm’s position on the patient’s side is essential for the success of remote robotic surgery in Tactile Internet (TI) environments. This paper presents a prediction model based on the Transformer-based Informer framework for accurate and efficient position estimation. The proposed method has combined with a Four-State Hidden Markov Model  to simulate realistic packet loss scenarios. The proposed approach addresses challenges such as network delays, jitter, and packet loss to ensure reliable and precise operation in remote surgical applications. The method also integrates the optimization problem into the Informer model by embedding constraints such as energy efficiency, smoothness, and robustness into its training process using a differentiable optimization layer.  

The topic studied in this paper is very interesting, and the innovation in theoretical methods is quite obvious. However, the following issues need to be further improved:

  1. The fifth and sixth innovation points should not be considered as innovative content. It is recommended to delete them.
  2. There is insufficient discussion in the section of "Related Work". Regarding the related challenging issues of remote robotic surgery, the current research status, and theoretical methods of the same type, more literature needs to be added to expand the discussion.
  3. In the section of "Optimization Problem", proofs for the solvability and convergence of the model need to be added.
  4. In the section of "Experimental Setup & Results", ablation experiments for verifying the AI model need to be further added.
  5. It is a pity that the method in this paper lacks real experiments. It is recommended to add experiments on the application of real robots.
  6. It is recommended to add the experimental conditions for the comparison methods in Tables 1 and 2. How were the experimental data obtained?
  7. It is recommended to add a discussion on the advantages and disadvantages of the method in this paper, as well as future research plans.

Author Response

Comment 1: The fifth and sixth innovation points should not be considered as innovative content. It is recommended to delete them.

Response 1: Acknowledged. We have removed the fifth and sixth points from the list of core contributions. Evaluation using the JIGSAWS dataset and model comparisons are now described separately after the contributions, rather than being listed as innovations.
(Updated in Section 1, Page 2, Lines 83–85)

Comment 2: There is insufficient discussion in the section of "Related Work". Regarding the related challenging issues of remote robotic surgery, the current research status, and theoretical methods of the same type, more literature needs to be added to expand the discussion.

Response 2: The Related Work section has been expanded to include recent studies covering network challenges, packet loss recovery, latency constraints in telesurgery, and predictive modeling for remote robotic systems.
(Expanded in Section 2, Pages 3-4, Lines 115–146.)

Comment 3: In the section of "Optimization Problem", proofs for the solvability and convergence of the model need to be added.

Response 3: A new subsection, Section 5.6 (Page 13, Lines 420439), titled "Solvability and Convergence Analysis," has been added. It includes two short theoretical proofs:

  • Proof of the existence and uniqueness of the solution based on convexity,

  • Proof of convergence using standard gradient descent theory.

Comment 4: In the section of "Experimental Setup & Results", ablation experiments for verifying the AI model need to be further added.

Response 4:Thank you for the valuable suggestion.
Ablation experiments were not included in this version of the study, as the primary focus was to validate the overall framework under simulated network impairments. We agree that ablation studies are important for analyzing the contribution of individual components (e.g., attention mechanism, optimization layer) to the final performance.

A note has been added to the Conclusion section (Page 18, Line 539) indicating that future work will include targeted ablation experiments to further strengthen the analysis and interpretability of the model.

Additionally, future work will include ablation studies to evaluate the contribution of different components of the proposed framework.”

Comment 5: It is a pity that the method in this paper lacks real experiments. It is recommended to add experiments on the application of real robots.

Response 5: We agree with the reviewer that validating the framework on a real robotic platform is a crucial next step. Although real robot testing was not feasible during this phase due to hardware limitations, we have updated the conclusion (Page 18, Lines 542–548) to clearly state our intent to perform physical experiments in future work.

In addition, a formal request has been submitted to the Intuitive Surgical Foundation to access the da Vinci Research Kit (dVRK), which will enable us to test the framework under real-time network conditions using an actual robotic arm in our lab environment.

Related Revised Text (Page 18, Lines 544–549):

"Although the study demonstrates strong performance in simulated environments, the value of real-world validation is fully recognized. In future work, the aim is to deploy the proposed framework on a physical robotic platform to assess its real-time effectiveness under actual network conditions. This step will help translate the predictive model from theory to practical impact in robotic-assisted surgical systems.”

Comment 6: It is recommended to add the experimental conditions for the comparison methods in Tables 1 and 2. How were the experimental data obtained?

Response 6: Additional details have been added at the end of Section 6.3.2 (Page 17, Lines 518–522), we clarified that:

  • All models were trained and tested on the same subset of the JIGSAWS dataset,

  • Separate kinematic data files were used for training and testing,

  • The same packet loss patterns (using a 4-state HMM) were applied uniformly,

  • Identical evaluation settings were used to ensure a fair and consistent comparison.

Comment 7: It is recommended to add a discussion on the advantages and disadvantages of the method in this paper, as well as future research plans.

Response 7: This has been addressed in the Conclusion section (Page 18, Lines 542–548). We summarized the key advantages, including high prediction accuracy under burst packet loss and low computational complexity. The main limitation (slightly lower Z-axis accuracy) is also acknowledged. Future plans include deploying the model on a real robotic system and performing ablation studies. (Page 18, Lines 539–541).

Reviewer 5 Report

Comments and Suggestions for Authors

The article is very interesting and timely.

Here are a few recommendations that I believe would improve it and boost its visibility:

1 Introduction
It would be useful to present a broader definition of the Tactile Internet, and to illustrate when this term first began to be discussed.

2 Related Works
Including a comparative table of the characteristics of the models analyzed would be very helpful.

4 Proposed Model
A schematic figure clarifying the HMM logic and the information flow would be useful.

6 Experimental Setup and Results
It would be interesting to add a logical diagram of the program implemented and tested in Python. This is, in fact, the core of the work, and it would be valuable to understand how everything functions.

Author Response

Comment 1: Introduction
It would be useful to present a broader definition of the Tactile Internet, and to illustrate when this term first began to be discussed.

Response 1: Thank you for the suggestion. The Introduction has been updated to include a broader definition of the Tactile Internet, describing its goal of enabling real-time interactive systems with haptic feedback, and highlighting its first formal discussion around 2014 by the IEEE.
This addition provides historical context and clarifies the relevance of the Tactile Internet to remote robotic surgery.
(Updated in Section 1, Page 1, Lines 23-30)

Comment 2: Related Works
Including a comparative table of the characteristics of the models analyzed would be very helpful.

Response 2: A comparative table summarizing the characteristics of the analyzed models (Informer, LSTM, RNN, and TCN) has been added to the Related Work section. The table highlights each model's complexity, handling of sequence data, suitability for long sequences, and key strengths.
(Added as Table 1, Page 4)

Comment 3: 4 Proposed Model
A schematic figure clarifying the HMM logic and the information flow would be useful.

Response 3: A schematic figure illustrating the logic of the 4-state Hidden Markov Model (HMM) and its integration into the Informer prediction framework has been added.
This diagram clarifies how the packet loss patterns are generated and injected into the data processing pipeline.
(Added in Section 4, Page 8, Figure 2)

Comment 4: 6 Experimental Setup and Results
It would be interesting to add a logical diagram of the program implemented and tested in Python. This is, in fact, the core of the work, and it would be valuable to understand how everything functions.

Response 4: Instead of a flow diagram, we have added Algorithm 1, which provides a clear, step-by-step outline of the Python-based implementation pipeline, from dataset preparation, HMM-based packet loss simulation, preprocessing, model training, to evaluation.
We believe this format provides a clear and reproducible description of the experimental process.
(Added in Section 6.2, Page 14)

Round 2

Reviewer 2 Report

Comments and Suggestions for Authors

The authors have provided reasonable responses to the concerns raised in the first review. Their justifications and clarifications made in the revised version address the issues highlighted in Report 1.

As a suggestion for future work, it would be valuable to consider real-world factors such as mechanical lag, actuator drift, and sensor noise, which could impact the system's performance in practical applications.

Reviewer 4 Report

Comments and Suggestions for Authors

I have no further comments.